# Unveiling the Pain Relief Potential: Harnessing Analgesic Peptides from Animal Venoms

**DOI:** 10.3390/pharmaceutics15122766

**Published:** 2023-12-13

**Authors:** Ana Flávia Marques Pereira, Joeliton S. Cavalcante, Davi Gomes Angstmam, Cayo Almeida, Gean S. Soares, Manuela B. Pucca, Rui Seabra Ferreira Junior

**Affiliations:** 1Center for the Study of Venoms and Venomous Animals (CEVAP), São Paulo State University (UNESP—Univ Estadual Paulista), Botucatu 01419-901, SP, Brazil; ana.f.pereira@unesp.br; 2Graduate Program in Tropical Diseases, Botucatu Medical School (FMB), São Paulo State University (UNESP—Univ Estadual Paulista), Botucatu 01419-901, SP, Brazil; joeliton.cavalcante@unesp.br (J.S.C.); davi.angstmam@unesp.br (D.G.A.); 3Center of Mathematics, Computing Sciences and Cognition, Federal University of ABC, Santo André 09280-560, SP, Brazil; cayoalmeida06@gmail.com; 4Delphina Rinaldi Abdel Azil Hospital and Emergency Room (HPSDRAA), Manaus 69093-415, AM, Brazil; geansoarex@outlook.com; 5Department of Clinical Analysis, School of Pharmaceutical Sciences, São Paulo State University, Araraquara 14801-320, SP, Brazil; manuela.pucca@unesp.br; 6Center for Translational Science and Development of Biopharmaceuticals FAPESP/CEVAP, São Paulo State University (UNESP—Univ Estadual Paulista), Botucatu 01419-901, SP, Brazil

**Keywords:** analgesic peptides, animal venoms, biomolecules, pain treatment, toxins

## Abstract

The concept of pain encompasses a complex interplay of sensory and emotional experiences associated with actual or potential tissue damage. Accurately describing and localizing pain, whether acute or chronic, mild or severe, poses a challenge due to its diverse manifestations. Understanding the underlying origins and mechanisms of these pain variations is crucial for effective management and pharmacological interventions. Derived from a wide spectrum of species, including snakes, arthropods, mollusks, and vertebrates, animal venoms have emerged as abundant repositories of potential biomolecules exhibiting analgesic properties across a broad spectrum of pain models. This review focuses on highlighting the most promising venom-derived toxins investigated as potential prototypes for analgesic drugs. The discussion further encompasses research prospects, challenges in advancing analgesics, and the practical application of venom-derived toxins. As the field continues its evolution, tapping into the latent potential of these natural bioactive compounds holds the key to pioneering approaches in pain management and treatment. Therefore, animal toxins present countless possibilities for treating pain caused by different diseases. The development of new analgesic drugs from toxins is one of the directions that therapy must follow, and it seems to be moving forward by recommending the composition of multimodal therapy to combat pain.

## 1. Introduction

The International Association for the Study of Pain provides a definition of pain as “an unpleasant sensory and emotional experience associated with, or similar to that associated with, actual or potential tissue damage” [1]. Pain can be either acute or chronic, mild or severe, constant or intermittent, throbbing or steady, and can be categorized as somatic or visceral. However, describing pain accurately, along with determining its precise location, is intricate, as it might be confined to a particular spot or encompass a widespread area. Comprehending the origins and mechanisms responsible for these pain variations is imperative for implementing proper management and pharmacological interventions. 

In this context, animal venoms have been demonstrated over the years to be potential sources of biomolecules that can be used as analgesic agents across a diverse range of pain models [2]. A vast array of animals, including annelids, cnidarians, echinoderms, mollusks, arthropods, and vertebrates, yield venoms, leading to a significant variability in their composition. While the potential harm posed by venomous animals is known, they have also been harnessed as therapeutic agents to treat a wide variety of inflammatory and infectious diseases [3,4,5].

This review aims to highlight the most promising venom-derived toxins that have been investigated as potential prototypes for analgesic drugs. Furthermore, it seeks to encourage scientists to explore untapped taxa for prospective analgesics and delve into the mechanisms of action of already identified molecules, some of which have been synthesized. The discussion extends to research perspectives and the obstacles to using toxins to obtain new analgesic molecules.

## 2. Pain Pathophysiology

Pain is a complex experience involving an actual or potential stimulus of something harmful and the physiological and emotional reactions to these events. Pain remains subjective, with each individual presenting the sensation uniquely through personal experiences. The individual then imparts unpleasant experiences across three distinct dimensions: (i) emotion, (ii) imagination, and (iii) sensation [6].

The noxious stimulus is detected by specialized peripheral sensory neurons known as nociceptors. These nociceptors have a dual projection—one extends as a peripheral axon to the skin and other organs, while the other projects toward the central nervous system (CNS) [7,8] (Figure 1). Following an injury, chemical mediators synthesized within the tissue undergo conversion into an electrical signal by transient receptor potential generator channels (TRP channels) and purinergic channels. Upon transduction, these neuronal events are transmitted by sensory fibers that project to Rexed’s lamina. Subsequently, from the spinal cord, the information travels to the brainstem and reaches the cerebral cortex, culminating in the perception of pain [9].

Physiological pain presents a protective purpose, yet it transforms into a pathological phenomenon in several instances (e.g., neuropathy and inflammation). Under these conditions, individuals and animals exhibit sensitivity to painful stimuli (hyperalgesia) [10], as well as responses to mechanical stimuli (mechanical allodynia) [11] and thermal (thermal allodynia) [12]. The pain can be categorized into nociceptive, neuropathic, and inflammatory when we considering pathological circumstances. In cases of physiological pain, the perception of it arises from nociception, the process through which noxious stimulation is relayed via the peripheral and central nervous systems.

### 2.1. Nociceptive Pain

Nociception is the response of the sensory nervous system to potentially harmful chemical, mechanical, and thermal stimuli, as well as actual or potentially harmful damage, triggered by the activation of nociceptors. Aδ and C fibers are the two types of nociceptors that have specialized free nerve endings and are widely located in the skin, muscles, joint capsules, bones, and some important internal organs [9].

### 2.2. Neuropathic Pain

Neuropathic pain is a type of chronic pain caused by damage to any level of the nervous system, either in peripheral regions, injuring peripheral fibers, or in regions of the central nervous system, such as the spinal cord and/or the brain [13]. Such diseases and/or injuries disrupt the normal activity of ion channels (sodium, calcium, and potassium), inhibitory interneurons, and descending modulatory control systems that modify the conduction of sensory signals to the spinal cord and the brain [14,15]. 

Common conditions associated with peripheral neuropathic pain include painful diabetic polyneuropathy [16], postherpetic neuralgia [17], trigeminal neuralgia [18], HIV-associated neuropathic pain [19], cancer-related pain [20], nerve compression (e.g., carpal tunnel syndrome herniated disc) [21], post-surgical pain (e.g., following mastectomy [22], herniorrhaphy [23], phantom limb pain [24], and others), and neuropathy from toxic exposure (e.g., antiretroviral agents, chemotherapeutics, and antituberculosis drugs) [25]. On the other hand, post-traumatic spinal cord pain, pain related to multiple sclerosis, post-stroke pain, transverse myelitis, post-radiation myelopathy, and HIV myelopathy are examples of central neuropathies.

### 2.3. Inflammatory Pain

Inflammatory pain arises as an inflammatory cascade is triggered, which then sensitizes peripheral nerve endings, with the aim of eliminating necrotic cells and initiating the process of tissue repair [26,27,28]. This type of pain can be classified as (i) acute inflammatory pain, which is usually intense and occurs in a short period of time; (ii) chronic inflammatory pain, which manifests with diverse levels of pain and persists beyond the period of healing [29]. After injury or damage, nociceptor-modulated peripheral sensitization occurs, causing primary hyperalgesia (exaggerated sensitivity to pain), or primary allodynia (sensation of pain when the stimulus is not normally painful). Chemical mediators such as histamine, bradykinin, acids, and serotonin are released and have the capacity to activate these nociceptors, making them depolarized or sensitized. Consequently, nociceptor terminals become more receptive to the same concentration of chemical mediators, enhancing their sensitivity [30,31]. 

Central sensitization is the amplification of pain by central nervous system mechanisms, which occurs independently of peripheral injury or inflammation. Repetitive and usually high-intensity synaptic transmission activates intracellular signal transduction cascades in dorsal horn neurons that enhance response to subsequent stimuli [32,33]. Several neurotransmitters participate in the perception of pain. This encompasses major types of both inflammatory and non-inflammatory mediators, second messenger production, their interaction with different channel-linked receptors, and the ensuing pharmacological effects at the terminals of both pre- and post-synaptic neurons (Figure 2). AMPA, NMDA, and metabotropic glutamate receptors, as well as the substance P (neurokinin) receptor NK1, and the BDNF (neurotrophin) receptor TrkB expressed by dorsal horn neurons, are involved in the induction of central sensitization [6]. In pre-synaptic sites, the activation of cytokine receptors and chemokine receptors results in phosphorylation and the activation of MAPK family members (ERK and p38), leading to the release of glutamate from synaptic vesicles via the activation of ion channels [34]. On the other hand, at post-synaptic sites there is an increase in neurotransmitters’ release (e.g., glutamate). When occur this release, prompts the activation of the MAPK family, which, in turn, can lead to central sensitization by upregulating NMDA (NMDAR) and AMPA (AMPAR) receptors downregulating potassium channels [9].

## 3. Animal Venoms: Composition and Effects

Due to advances in the development of analytical tools, numerous venom-producing animal species, including snakes [35], scorpions [36], spiders [37], bees [38,39,40], frogs [41], and mollusks [42,43,44], among other animals, have their venoms studied in terms of the composition and identification of toxins and peptides, providing an overview of the composition of these venoms. This allowed us to understand that venoms are multifunctional and complex systems, comprising proteins with and without enzymatic activities, peptides, and amino acids, among other components.

Snake venoms are true mutable substances, composed of almost unique formulations [45,46,47,48,49], featuring in its composition mainly proteins, peptides, and their respective isoforms. These molecules are distributed in dominant families: secreted A2 phospholipases (PLA2s), snake venom metalloproteinases (SVMPs), snake venom serine proteases (SVSPs), and three-finger peptides (3FTXs). Also we can cite secondary secretory proteins, kunitz peptides, L-amino acid oxidase (LAAO), the cysteine-rich secretory protein (CRiSP), C-type lectins and C-type lectin-like (CTL/SNACLEC), disintegrin (DISI) and natriuretic peptides (NPs), including vasoactive peptides, bradykinin potentiating peptides, and inhibitory peptides [35].

Some venoms are able to induce pain [6,42], edema, local and systemic inflammatory process [50,51,52], oxidative stress [53], the functional destabilization of the coagulation system [54,55], acute kidney injury [56], rhabdomyolysis, and necrosis. In addition, other snake venoms can trigger a neurotoxic syndrome: flaccid paralysis (bilateral ptosis and ophthalmoplegia) and flaccid neuromuscular paralysis (descending), which can be aggravated by involving the bulbar blockade (mouth and throat muscles responsible for speech and swallowing) and respiratory muscles [57].

Scorpion venoms are characterized by various peptides that are responsible for triggering physiopathological manifestations upon envenomation. Following envenomation, these peptides interact with ion channels, while the enzymes present in the venom facilitate the biodistribution of venom components [58]. The components of scorpion venoms and their encoding transcripts are classified into eight main categories: (i) Na^+^ channel toxins, (ii) K^+^ channel toxins, (iii) Ca^2+^ channel toxins, (iv) Cl^−^ channel toxins, (v) TRP channel toxins, (vi) enzymes, (vii) protease inhibitors, and (viii) defense peptides [36]. During envenomation by Buthidae scorpions, clinical manifestations range from localized signs, systemic manifestations, and vital insufficiency to dangerous cardiorespiratory symptoms, often including cardiogenic shock, pulmonary edema, or severe neurological impairment (coma and/or convulsions) [59].

The use of venom is also a success story for spider predation purposes due to its effectiveness against a wide variety of prey and in defense. Although the effect of spider venom toxins on humans is overestimated, bites that result in severe envenomation are limited to a restricted number of spider taxa and include mygalomorph Australian funnel-web spiders (*Atrax* and *Hadronyche*, *Atracidae*), the araneomorph recluse spiders (*Loxosceles*, Si-cariidae), widow spiders (Latrodectus sp., Theridiidae), and wandering spiders (*Phoneutria*, *Ctenidae*). The ability to induce pathophysiological manifestations in humans also directly influences research on the venom. Therefore, the composition of spider venom was for a long time largely restricted to specific taxa that have public health relevance. In general, spider venoms are divided into four groups: (i) small-molecular-mass compounds (SMMSs) which include ions, organic acids, nucleotides, nucleosides, amino acids, amines, and polyamines; (ii) antimicrobial peptides (cytolytic or cationic) that are found in only a few families of spiders; (iii) peptide neurotoxins that modulate a wide variety of channels and receptors present in the membranes of excitable cells (for example, nerves and muscles); and (iv) proteins and enzymes, which include cysteine-rich secretory proteins (CRISPs), hyaluronidases, collagenases, and phospholipases. These, in turn, act on the extracellular matrix or cell membrane, facilitating the movement of toxins into the prey as so-called dissemination factors [37].

Bee venom is a complex mixture of compounds, which include proteins, peptides, amino acids, phospholipids, sugars, biogenic amines, volatile compounds, pheromones, and a large amount of water (>80%). We can also find present in the venom, adrenaline, dopamine, histamine, hyaluronidase, noradrenaline, phospholipases A 2 (PLA2s), phospholipases B (PLBs), serotonin, apamine, melittin, and mast cell degranulator peptide (MCD) [38,39,40]. Honeybee envenomation may cause anaphylactic shock, and in multiple stings, entire organ systems can be affected, cause cardiovascular diseases, pulmonary dysfunction, renal failure, hepatic damage, and hematologic diseases [60].

Although there is no apparent clinical relevance to human health, frog poisonings are commonly reported in domestic animals, highlighting their veterinary importance. The venoms secreted by the parotid glands of frogs correspond to a mixture of biologically active compounds, including dopamine, epinephrine, norepinephrine, serotonin, bufotenin, bufagenins, bufotoxins, and indolealkylamines, which are rapidly absorbed by the mucous membranes (including the gastric mucosa, in the cases of ingestion). Upon entering the systemic circulation, the greatest effects are seen in the heart, the peripheral vasculature, and the nervous system [41,61]

The venomous mollusks from the superfamily Conoidea have venoms with 100 to 200 components in their composition. There are small peptides conotoxins of 5–50 amino acids that have high biological activity. These peptides are classified according to number of cysteine residues and pharmacological targets. The most extensively researched conotoxins include α, ω, κ, μ, and δ toxins, that acts inhibiting nicotinic acetylcholine receptors, voltage-gated calcium channels, voltage-gated potassium channels, voltage-gated sodium channels, and voltage-gated sodium channels, respectively. The *Conus* species envenomation may be fatal, causing immobilization and the inhibition of neuromuscular transmission, leading to cardiac and neurological disorders [42,43,44].

Animal venoms have shown a high potential for the development of products, some of which are already used clinically [62]. This is due to the range of molecules with different biological functions, which make animal venoms a true source for the mining of new compounds with potential, both for use in basic research and for the development of new drugs (antitumor, antimicrobial, antiviral, and anti-inflammatory drugs, among others). In contrast, peptide toxins are generally found in lesser abundance; however, their diversity as well as ease of synthesis have attracted substantial attention. Despite their diversity, venom peptides appear to have evolved from a relatively small number of structural structures that are particularly suited to addressing the crucial issues of potency and stability, which makes them a unique source of linkages and structural models important to build new therapeutic agents.

## 4. Toxins Targeting Pain: Discovering Potential Analgesics

During pain signaling, nociceptors and post-synaptic neurons are activated or inhibited by neurotransmitters binding to many channels and transporters that contribute to the release of pain-related neurotransmitters and/or ion fluxes relevant to stimulus conduction. Pain can be modulated by venom-derived toxins, such as by selective toxins that bind to receptors responsible for modulating the detection, conduction, and propagation of painful impulses, enabling the use of these toxins in the design and development of novel analgesic drugs. Despite the variety of pharmacological targets, ion channels have drawn attention. Voltage-gated sodium channels (VGSCs) and transient receptor potential (TRP) channels are crucial therapeutic targets in pain management, which form the critical components of the nociceptive sensory pathway.

The initial investigations reporting the analgesic activities of animal venoms primarily focused on snake venoms, with some studies mentioning bee venom. From 1970 to 2004, there was an exponential increase in the frequency of published articles that investigated venoms and toxins in relation to nociception. After 2004, the publication trend diverged from the previously observed pattern of pain-related articles, reaching a plateau that resulted in a linear progression in the field of venom and toxin research [63]. This change in trajectory could potentially be attributed to the discovery of novel venoms, along with the development of methodologies for evaluating their effectiveness. Therefore, we conducted a contemporary analysis, meticulously selecting bioprospecting investigations involving analgesic peptides examined in animal models.

### 4.1. Snake Analgesic Peptides

Several toxins purified and synthesized from snake venom have biotechnological potential for the treatment of various human diseases [64,65,66,67]. As far as the search for new analgesic drugs is concerned, snake venoms have been little explored, perhaps due to their widely reported potential for inducing algesia in human envenomations. However, crotoxin (CTX) isolated from *Crotalus* spp. venoms, Najanalgesin and Cobra neurotoxin from *Naja naja atra* venom, Mambalgin-1 and Mambalgin-3 from *Dendroaspis polylepis polylepis* venom [68], µ-EPTX-Na1 from *D. polylepis polylepis* venom, and µ-EPTX-Na1 from *Naja attracts* [69] have demonstrated a variety of analgesic activities.

CTX induces analgesia via formyl peptide, α2-adrenergic, and muscarinic receptors in the acute. Also we can observe in chronic phases of hypernociception induced via partial sciatic nerve ligation [70]. It is dependent on formyl peptide, lipoxygenase, and muscarinic receptors, as shown in MOG 35-55-induced experimental autoimmune encephalomyelitis, an animal model of multiple sclerosis [71]. The previously outlined crotalfine, described in earlier years, has recently demonstrated its ability to intervene in mitigating the desensitization process of TRPA1 receptors [72]. This result in the induction of analgesia in pain protocols, specifically within the contexts of cold hypersensitivity through ciguatoxin-induced expression and and also mechanical hypersensitivity provoked by bradykinin- and zymosan-induced expression [73]. Najanalgesin and cobra neurotoxin induce analgesia through adenosine receptor (A1 and A2A) pathway activation in acute pain models induced by hot plate and spinal cord injury [74] and the c-Jun N-terminal kinase (JNK) inhibitor in neuropathic pain induced by spinal nerve ligation [75]. However, the induction of analgesia promoted by µ-EPTX-Na1 involves the Nav1.8 channel inhibitor [69], while mabalgin 1 and 3 present a common analgesia mechanism via acid-sensing ion channel (ASIC) inhibitors, as evidenced by the motor behavior tests such as the accelerated rotarod test and the grip strength test. This phenomenon can be observed in pain and inflammation models induced by carrageenan and thermal or mechanical von Frey tests. Furthermore, the isolated Cobratoxin from the venom of *Naja naja kaouthia* has exhibited analgesic effects in acute pain models targeting the α7 subunit of nicotinic acetylcholine receptors when administered intrathecally, thereby eliciting an antinociceptive response in mice [76] (Table 1).

### 4.2. Scorpion Analgesic Peptides

Scorpionism is a major public health problem in subtropical areas worldwide even though less than 25 species are considered dangerous to humans. The “Old World” (Africa) species of medical interest belong to the genera *Andoctonus*, *Buthus*, *Hottentota*, and *Leiurus*, while the “New World” (America) species belong to the *Centruroides* and *Tityus* genera, belonging to the Buthid family. One of the most prominent clinical signs following a scorpion sting is pain, which is mainly caused by neurotoxins targeting ion channels. Remarkably, the same scorpion venoms able to induce pain mediated by several mechanisms also exhibited analgesic activities, thereby establishing the groundwork for a potential pain-relieving agent.

Different toxins from the scorpion *Buthus martensii* are promising for the treatment of pain in a xenograft tumor mouse model [77,78]. These toxins can be important candidates also to use in acute thermal pain model induced by a hot plate, in acute inflammation model induced by formalin [79,80,81,82], in a mouse-twisting pain model, mechanical allodynia, in a nociception model induced by a shutter-controlled lamp, and in an inflammation model induced by formalin and acetic acid writhing [81]. Furthermore, TsNTxP isolated from *Tityus serrulatus* venom proved to be a good candidate in acute nociception induced by a water bath and a neuropathic pain model induced by the constriction of a sciatic nerve (CCI model) [83], while Leptucin from *Hemiscorpius lepturus* venom showed analgesic potential against an acute thermal pain model induced by a hot plate and a nociception model using the tail-flick test [84].

Peptides derived from scorpion venoms exhibit a wide variety of analgesic mechanisms: (i) ion channel inhibitors; (ii) MCF-7 and MDA-MB-231 migration inhibitors; (iii) MAPK inhibitors; (iv) arginine residue at position 58; and (v) CHis6-rAGAP NHis6-rAGAP. The various pharmacological mechanisms of scorpion venoms have likely drawn considerable attention to the bioprospecting due peptides present in this venom, while the venom of other species/genera remains underexplored and even unknown (Table 2).

### 4.3. Spider Analgesic Peptides

Spiders (order Araneae) currently comprise 49,483 species, almost all of which have the capacity to produce venom (WSC, 2021). However, in recent years, the exploration of spider venoms in search of prototype peptides for the design of new analgesic drugs has been restricted to a small number of genera. There is extensive research involving *Phoneutria nigriventer* peptides, some isolated from the venom, while others synthesized and/or obtained recombinantly. The venom of this spider species has represented an extensive shelf of compounds with potential for the development of new analgesic drugs in different pain models: inflammatory pain; cold/heat/mechanical hyperalgesia and allodynia; post-operative pain; hyperalgesia; neuropathic pain; cancer-related pain; mechanical pain; fibromyalgia; and others [87,88,89,90,91] (Table 3).

Peptides from *P. nigriventer* exhibit a wide variety of analgesic mechanisms, which include (i) a reduction in glutamate levels in the cerebrospinal fluid [92]; (ii) an effect on peripheral opioid and cannabinoid systems [93,94]; (iii) the activation of the NO-cGMP pathway -KATP [95]; (iv) the inhibition of spinal AChE resulting in the activation of muscarinic and nicotinic receptors [96]; (v) the selective antagonism of TRPA1 channels [92]; (vi) the reversible inhibition of voltage-gated calcium channels (VGCC) [97]; and (vii) the antagonism of the CXCR4 receptor [98]. There is a great diversity of pharmacological mechanisms in peptides of *Phoneutria nigriventer* venom, but the venom of other species/genera remains underexplored and even unknown.

The potential for the treatment of pain in underexplored spider venoms is mainly due to the inhibition of ion channels that control the transit of neurotransmitters. Furthermore, an interesting potential was reported by Deuis et al. [99] for μ-theraphotoxin-Pn3a isolated from the venom of *Pamphobeteus nigricolor*. The intraperitoneal injection of this peptide induces analgesia by inhibiting Nav1.7 channels and exhibited synergy with opioids in a model of peripheral pain induced by the injection of α-scorpion toxin OD1. Other peptides isolated from *Ceratogyrus* sp., *Davus* sp., *Haplopelma lividum*, *Heteropoda venatoria*, *Thrixopelma pruriens*, and *Grammostola porteni* also present potent candidates for analgesic drugs through the modulation of ion channels [79,87,88,90,91,96,99,100,101,102,103,104].

**Table 3 pharmaceutics-15-02766-t003:** Spider-derived toxins able to inhibit pain.

Toxin/ Molecule	Species	Production	Administration	Mechanism of Analgesia	Model	Ref.
PhTx3-4	*Phoneutria nigriventer*	Purified from venom	Intrathecal (i.t.) injection	Reduction in glutamate levels in the cerebrospinal fluid	Persistent inflammatory pain and post-operative (plantar incision) nociception in mice	[92]
PnPP-19	*Phoneutria nigriventer*	Synthesized	Subcutaneous (s.c.) injection	Both opioid and cannabinoid peripherical systems	Hyperalgesia induced by the administration of PGE2	[94]
Tx3-5	*Phoneutria nigriventer*	Purified from venom	Intrathecal (i.t.) injection	-	Postoperative (plantar incision), neuropathic (partial sciatic nerve ligation), and cancer-related pain (inoculation with melanoma cells) in animals	[105]
δ-CNTX-Pn1a	*Phoneutria nigriventer*	Purified from venom	Intrathecal (i.t.) injection	Opioid and cannabinoid systems	Hyperalgesia induced by the CCI model and hyperalgesia induced by the administration of PGE2	[93]
PnPP-19	*Phoneutria nigriventer*	Synthesized	Subcutaneous (s.c.) injection	Activation of NO-cGMP-KATP pathway	Hyperalgesia induced by the administration of PGE2	[95]
PhKv	*Phoneutria nigriventer*	Purified from a PhTx3 fraction	Intrathecal (i.t.) injection	Inhibition of spinal AChE resulting in the activation of the muscarinic and nicotinic receptors	Chronic constriction injury model and after intraplantar injection of capsaicin	[97]
Phα1β	*Phoneutria nigriventer*	Purified from venom	Intrathecal (i.t.) or intraplantarly (i.pl.) injections	Selective antagonist of TRPA1 channels	Nocifensive responses evoked by reactive TRPA1 channel agonist, mechanical and cold hyperalgesia, neuropathic pain induced by the chemotherapeutic agent bortezomib	[106]
Pha1β	*Phoneutria nigriventer*	Purified from venom	Intrathecal (i.t.) injection	Reversibly inhibits the voltage-gated calcium channels (VGCC)	Nociception that was triggered by capsaicin, CCI model, and hyperalgesia was induced in the melanoma cancer pain model	[97]
Pha1β	*Phoneutria nigriventer*	Recombinant	Intrathecal (i.t.) injection	Reversibly inhibits the voltage-gated calcium channels (VGCC)	Nociception that was triggered by capsaicin, CCI model, and hyperalgesia was induced in the melanoma cancer pain model	[97]
PnTx4(5-5)	*Phoneutria nigriventer*	Purified from venom	Intraplantarly (i.pl.) and subcutaneous (s.c.) injection	The antinociceptive effect of PnTx4(5-5) can also be related to the glutamatergic system	Hyperalgesia induced by PGE2, carrageenan, and L-glutamate (L-Glu)	[107]
Phα1β	*Phoneutria nigriventer*	Recombinant	Intravenous (i.v.) injection	-	Pain was induced by the CCI model and paclitaxel-induced acute and chronic pain	[108]
CTK 01512-2	*Phoneutria nigriventer*	Recombinant	Intrathecal (i.t.) injection	Reversibly, and not specifically a block of Cav 2.2 that affects the intracellular Ca^2+^ influx and the glutamate release	Mechanical and thermal hyperalgesia and cold allodynia	[109]
Phα1β	*Phoneutria nigriventer*	Purified from venom	Intrathecal (i.t.) injection	Blocker voltage-dependent calcium channels and the antagonism with the receptor CXCR4	Diabetic neuropathic pain	[98]
PnAn13	*Phoneutria nigriventer*	Synthetic peptide	Intrathecal (i.t.) injection	Opioid and cannabinoid systems	Hyperalgesia induced by PGE2	[110]
Phα1β	*Phoneutria nigriventer*	-	Intrathecal (i.t.) injection	Blocker TRPA1 and Cav2.2 receptors	Postoperative (plantar incision)	[111]
Phα1β	*Phoneutria nigriventer*	Recombinant	Intravenous (i.v.) injection	High-voltage calcium channel inhibitors (HVCCs) and cationic channel antagonists of the potential transient receptor (TRPA1)	Hyperalgesia and mechanical allodynia induced by reserpine (fibromyalgia)	[112]
Cd1a	*Ceratogyrus darlingi*	Chemical synthesis	Intraplantar (i.pl.) injection	Nav/Cav channel inhibitor	Peripheral pain by NaV channels induced with α-scorpion toxin OD1	[87]
JZTX-X	*Chilobrachys jingzhao*	Chemical synthesis	Intraplantar (i.pl.) injection	Kv4 channel inhibitor	Mechanical hypersensitivity	[100]
μ-TRTX-Ca2a	*Cyriopagopus albostriatus*	Purified from venom	Intrathecal (i.t.) and intraplantar (i.pl.) injection	Na v 1.7 channel inhibitor	Acute inflammation models induced by formalin and acetic acid and acute thermal pain models induced by hot plate	[101]
µ-TRTX-Ca1a	*Cyriopagopus albostriatus*	Purified from venom	Intraperitoneal (i.p.) injection	hNa v 1.7 channel inhibitor	Pain formalin-induced paw licking, hot plate test, and acetic acid-induced writhing	[102]
CyrTx-1a	*Cyriopagopus schioedtei*	Purified from venom	Intraperitoneal (i.p.) injection	hNa V 1.7 channel inhibitor	Thermal hyperalgesia	[103]
Df1a-NH2	*Davus fasciatus*	Chemical synthesis	Intraplantar (i.pl.) injection	NaV and CaV3 channel inhibitor	Peripheral pain by NaV channels induced with α-scorpion toxin OD1	[88]
Df1a-OH	*Davus fasciatus*	Chemical synthesis	Intraplantar (i.pl.) injection	NaV and CaV3 channel inhibitor	Peripheral pain by NaV channels induced with α-scorpion toxin OD1	[88]
GpTx-1 and GpTx-1-71	*Grammostola porteni*	Chemical synthesis	Intrathecal (i.t.) and Intracerebroventricular (i.c.v.) injection	Nav channel inhibitor	Tail-flick test, carrageenan- or complete Freund’s adjuvant (CFA)-induced inflammatory pain model, neuropathic pain model, mechanical allodynia, thermal hyperalgesia, writhing test, formalin test, tolerance evaluation, rotarod test, open field test, gastrointestinal transit test	[104]
HpTx3	*Heteropoda venatoria*	Purified from venom	Intramuscular (im.i) and intraperitoneal (i.p.) injections	Nav1.7 channel inhibitor	Acute inflammation models induced by formalin and acetic acid, chronic inflammation pain models induced by complete Freund’s adjuvant, acute thermal pain models induced by hot plate, and chronic neuropathic pain models induced by spared nerve injury	[90]
μ-theraphotoxin-Pn3a	*Pamphobeteus nigricolor*	Purified from venom	Intraperitoneal (i.p.) injection	Nav1.7 channel inhibitor and synergy with opioids	Peripheral pain by NaV channels induced with α-scorpion toxin OD1	[99]
ProTx-III	*Thrixopelma pruriens*	Recombinant	Intraplantar (i.pl.) injection	Nav1.7 channel inhibitor	Peripheral pain by NaV channels induced with α-scorpion toxin OD1	[91]
ProTx-II (β/ω-terafotoxina-Tp2a)	*Thrixopelma pruriens*	-	Intracerebroventricular (i.cv) injection	Nav1.7 channel inhibitor	Acute thermal pain model induced by hot plate and mechanical allodynia	[96]

### 4.4. Frogs Analgesic Peptides

Little is known about anuran venoms, which is partly due to the low number of poisonings with these animals. Anurans have granular glands that produce venoms composed of peptides, biogenic amines, steroids, and alkaloids, which induce cardiotoxic, neurotoxic, myotoxic, anesthetic, antimicrobial, hemolytic, healing, antidiabetic, and other effects [61,113]. Although the potential of the venoms of these animals is recognized, only the peptide bufalin, purified from the venom of *Bufo gargarizans*, was studied as an analgesic agent in an animal model [114]. Intraperitoneally, bufaline was able to inhibit the pain caused via the injection of formalin and carrageenan and presented pharmacological potential against mechanical and thermal allodynia via the inhibition of the activities of VGSCs [114] (Table 4). 

**Table 4 pharmaceutics-15-02766-t004:** Frog-derived toxins able to inhibit pain.

Toxin/ Molecule	Species	Production	Administration	Mechanism of Analgesia	Model	Ref.
Bufalin	*Bufo gargarizans*	Purified from venom	Intraperitoneal (i.p.) injection	VGSC activity inhibitor	Pain formalin-induced paw licking, pain carrageenan-induced thermal and mechanical hyperalgesia	[114]

### 4.5. Bee Analgesic Peptides

The *Apis mellifera* venom has peptides, bioactive compounds, and proteins in its composition. The main components are melittin (most abundant representing 40–60% of the dry weight of the venom), phospholipase A2 (bvPLA2), and apamin (a peptide). Some properties of this venom have already been reported in the literature, such as its anticancer, antinociceptive, and anti-inflammatory properties [115]. Bioprospecting studies has been carried out with the raw venom of *A. mellifera* bees to show the ability to induce analgesia in pain caused by spinal cord injury (SCI)-induced allodynia; thermal hyperalgesia [116]; oxaliplatin-induced mechanical allodynia [117]; complex regional pain syndrome type 1 (CRPS-I)-induced mechanical allodynia [118]; oxaliplatin-induced neuropathic pain, cold and mechanical allodynia [119]; vincristine-induced cold and mechanical hypersensitivity [120]; scalding-burn-model-induced mechanical allodynia [121]; and paclitaxel-induced cold and mechanical allodynia [120,122] (Table 5).

*A. mellifera* venom and its toxins cause analgesia through (i) effects on α-1 adrenergic and α-2 adrenergic receptors [116,120,122,123]; (ii) serotonergic (5-HT1/5-HT2 and 5-HT3) and opioid receptors [120]; (iii) the inhibition of COX expression [118,124]; (iv) the suppression of glial cells [121]; (v) the suppression of NK-1 receptor expression in mice with complex regional pain syndrome type I (CRPS I) [118]; (vi) an increase in the A-fiber action potential threshold of DGR neurons [119]; and (vii) the inhibition of the expression of substance P in the central and peripheral nervous system [121].

**Table 5 pharmaceutics-15-02766-t005:** Bee venom able to inhibit pain.

Toxin/ Molecule	Species	Production	Administration	Mechanism of Analgesia	Model	Ref.
Bee venom	*Apis mellifera*	Sigma-Aldrich^®^	Subcutaneous (i.s.) injection	Suppression of glial cell activation ipsilateral, dorsal spinal cord	Spinal cord injury (SCI)-induced allodynia and thermal hyperalgesia	[116]
Bee venom	*Apis mellifera*	Sigma-Aldrich^®^	Subcutaneous (i.s.) injection	α-2 adrenergic receptors activation	Oxaliplatin-induced mechanical allodynia	[117]
Bee venom	*Apis mellifera*	-	Subcutaneous (i.s.) injection	Decrease of NK-1 receptor expression in dorsal root ganglia (DRG)	Complex regional pain syndrome type-1 (CRPS-I) induced mechanical allodynia	[118]
Bee venom	*Apis mellifera*	Jayeonsaeng TJ^®^	Subcutaneous (i.s.) injection	Increase of the action potential threshold in A-fiber DRG neurons	Oxaliplatin-induced neuropathic pain, cold and mechanical allodynia	[119]
Bee venom	*Apis mellifera*	Jayeonsaeng TJ^®^	Subcutaneous (i.s.) injection	α-2 adrenergic receptors activation with the involvement of noradrenergic nuclei of the locus coeruleus (LC)	Vincristine-induced cold and mechanical hypersensitivity	[120]
Bee venom	*Apis mellifera*	Apis Injeel^®^	Intraperitoneal (i.p.) Injection	Inhibitory activity of the COX pathway	Complete Freund’s adjuvant-induced arthritic rats	[124]
Bee venom	*Apis mellifera*	Sigma-Aldrich^®^	Subcutaneous (i.s.) injection	Inhibitory effect on the expression of substance P in the peripheral and central nervous systems	Scalding-burn-model-induced mechanical allodynia	[121]
Bee venom + Morphine	Bee venom from *Apis mellifera*	Sigma-Aldrich^®^	Intrathecal (i.t.) injection	Spinal opioidergic and 5-HT3 receptors modulate the analgesia	Oxaliplatin-induced neuropathic pain, cold and mechanical allodynia	[125]
Bee venom + Venlafaxine	Bee venom from *Apis mellifera*	Jayeonsaeng TJ^®^	Intrathecal (i.t.) injection	α-2 adrenergic receptors activation and serotonergic receptors (5-HT1/5-HT2 and 5-HT3)	Paclitaxel-induced cold and mechanical allodynia	[126]
Bee venom and melittin	*Apis mellifera*	Purified from bee venom	Subcutaneous (i.s.) injection	α-2 adrenergic receptor activation	Paclitaxel-induced mechanical hyperalgesia	[122]
Melittin	*Apis mellifera*	Sigma Aldrich^®^	Intraprostatic injection	Suppression of COX-2 expression	Complete Freund’s adjuvant-induced prostatitis	[127]
Melittin	*Apis mellifera*	Purified from bee venom—obtained from Sigma-Aldrich^®^	Subcutaneous (i.s.) injection	Spinal α-1 and α-2 adrenergic receptor activation	Oxaliplatin-induced mechanical and cold allodynia	[128]
Phospholipase A2 (bvPLA2)	*Apis mellifera*	Sigma-Aldrich^®^	Intraperitoneal (i.p.) injection	Activation of the noradrenergic system, via α2-adrenergic receptors	Oxaliplatin-induced neuropathic pain—cold and mechanical allodynia	[123]
bvPLA2	*Apis mellifera*	Sigma-Aldrich^®^	Intraperitoneal (i.p.) injection	Suppressing immune responses in the DRG by regulatory T cells (Tregs)	Oxaliplatin-induced neuropathic pain in Treg-depleted mice	[128]

### 4.6. Mollusk Analgesic Peptides

Conotoxins or conopeptides are small peptides presenting from 10 to 35 amino acids obtained from the venom of cone snails. They encompass a wide range of bioactive compounds, including those with analgesic potential, as is the case of the medicine Ziconotide (Prialt^®^), which is derived from a *Conus magus* conotoxin (ω-conotopeptide MVIIA) and is used for the treatment of chronic pain [129,130]. There are other marine snails with potential venoms, which are distributed in three families, namely Conoidea (containing the *Conus* genus), Terebridae (these are the “auger snails”), and Turridae (composed of the “turrids”), although the most studied and known peptides belong to the *Conus* genus [129].

Mollusk venoms have also been studied and represent a truly rich source of bioactive peptides. Among their diverse effects, their analgesic property has already been evidenced in pain models such as the rat NLP model [131] and the hot-plate model [132], as well as in rat partial sciatic nerve injury (PNL model), chronic constriction injury pain (CCI model) [133,134,135,136], chronic visceral hypersensitivity (CVH model) [137], oxaliplatin-induced neuropathic pain cold and mechanical allodynia [138], paclitaxel-induced neuropathic pain [139], neuropathy and oxaliplatin-induced neuropathic pain [140,141], post-surgical pain (PSP model), and cisplatin-induced pain [141].

Mollusk peptides act mainly via the inhibition of high-voltage-activated N-type calcium channel currents (Cav2.2) in isolated mouse dorsal root ganglia (DRG) neurons and the inhibition of G protein-coupled γ-aminobutyric acid type B receptor (GABABR)-coupled Cav2.2 channels in rat DRG neurons, which results in a reduction in the excitability of DRG neurons [134,135,141,142]. In addition, other mechanisms through which mollusk peptides induce analgesia have already been reported, such as the inhibition of TTX (tetrodotoxin)-resistant sodium currents in DRG neurons [132], the inhibition of NR2B ion channels [141], the inhibition of α9-containing nicotinic acetylcholine receptors (nAChRs) [143], and the activation of GABABRs [136]. However, the analgesia mechanism of other peptides still needs to be elucidated (Table 6).

## 5. Challenges in Integrating Venom-Derived Toxins for Pharmaceutical Pain Relief Solutions

When it comes to developing new biopharmaceuticals derived from venom toxins, two fundamental challenges must be addressed to effectively position these compounds as viable candidates in the pharmaceutical market. The first challenge revolves around determining the most suitable mode of administration for these compounds. How can these molecules efficiently reach their intended targets using the least invasive route? 

For instance, let us examine the drug Prialt^®^ (Ziconotide), which is primarily administered intrathecally to patients [150]. Despite the inherent advantages of this approach, such as reducing complications associated with catheters and significantly enhancing pain control efficacy, especially in advanced stages of carcinomas, this method of administration still qualifies as invasive, similar to parenteral routes like intravenous, subcutaneous, and intramuscular injections [151,152,153]. The desired goal is to adopt a route that minimizes harm to the organism and results in a lower incidence of adverse reactions. Among the considered options are oral, nasal, pulmonary, and rectal routes [151,154].

The second inherent challenge in pharmaceutical development concerning peptides and proteins revolves around the search for molecules with low mass that still possess specific biological activities. Craik et al. [155] and Lipinski [156] contend that molecules with a molecular weight below 500 Da are amenable to oral administration, while those exceeding 5000 Da forfeit this suitability due to their size. However, larger molecules demonstrate high specificity toward their target receptors, whereas smaller ones lack selectivity and often result in more side effects. It is apparent that many of the peptides currently under investigation still possess significant dimensions, which impede their ability to reach target receptors. Consequently, this affects absorption, metabolism, and distribution, ultimately making the choice of administration route more complex. These challenges underscore the complexity of developing venom-based pharmaceuticals for pain relief and emphasize the ongoing need for research and innovative approaches in this area.

## 6. Future Perspectives of Using Toxins as Novel Analgesics

Despite the challenges associated with the pharmaceutical utilization of peptides, the industry has made significant strides. Currently, there are over 150 peptides in clinical development, with an additional 400 to 600 peptides in various stages of preclinical trials. Moreover, there are more than 80 peptide-derived drugs already available in the market, generating a worldwide revenue exceeding USD 20 billion just in 2019. These therapeutic peptides are indicated for a wide array of conditions, including but not limited to type 1 and 2 diabetes, cancer, neuropathic pain, hypertension, osteoporosis, cardiovascular diseases, HIV, and various endocrine and metabolic disorders [155,157].

To ensure the effective action of a biomolecule at its therapeutic target, it is essential to consider the drug delivery system. This field of study focuses on methods for encapsulating proteins and peptides with the precise aim of directing them to the desired site of action [151,155,157]. Several techniques are under development, each with its own specificities, such as the use of liposomes, the spray drying technique, solid lipid nanoparticles (SLNs), double emulsions, and other approaches [158,159,160,161,162]. The objective is to safeguard the biomolecule from degradation, enabling it to reach its destination within the body effectively and thus optimizing its therapeutic capacity. The drug delivery field continues to evolve through ongoing research efforts aimed at enhancing drug delivery technologies to make treatments more efficient and safer [151,163].

Due to this promising development of drugs based on peptides, future perspectives of using toxins as novel analgesics involve strategies for reducing costs for the large-scale synthesis and purification of peptides, considering reducing costs using chemical synthesis and expression of recombinant peptides [164,165]. Currently, one of the key challenges in the bioprospecting of drugs derived from animal venoms is the acquisition of validated venom batches produced in accordance with stringent Good Production Practices (GMP) [166]. This is a costly endeavor, demanding a substantial number of animals. In situations involving animals like spiders and scorpions, the quantity of venom obtained often falls short of meeting the requirements for all stages of discovery and preclinical testing. To address these challenges, alternative methods include chemoenzymatic peptide synthesis, which relies on enzyme-mediated peptide bond formation rather than chemical reagents. This approach offers cost reductions compared to solid-phase peptide synthesis (SPPS). Additionally, automated flow peptide synthesis (AFPS) is emerging as a swifter method than traditional SPPS [157].

As we look ahead, the development of innovative peptide-derived drugs is poised to benefit from fresh perspectives on peptide design. These perspectives encompass considerations of protein–protein interactions, integration with nanotechnology-based platforms such as nanoparticles and liposomes, conjugation with proteins, the lipidation or PEGylation of peptides, and structural modifications. These approaches hold promise for optimizing drug delivery; surmounting challenges related to renal clearance; and enhancing the biological activity, stability, and solubility of peptide-based therapeutics [155,157,167].

## 7. Conclusions

The venoms of spiders, mollusks, scorpions, bees, frogs, and snakes offer a rich source of new pharmaceutical compounds. These venoms contain peptides that play a fundamental role in defense and predation, and they can interfere with multiple vital biological processes. Within this context, the action of numerous compounds affecting rapid synaptic transmission has provided extensive research into their pharmacological potential for pain treatment. As plausible candidates in the development of analgesics, peptides from animal venoms show selectivity for receptors associated with pain modulation in different pathological models, with different mechanisms of analgesia, which makes them valuable for proof-of-concept studies and the development of novel analgesic drugs or associated with multimodal therapies. However, for therapeutic applications, it is necessary to address a number of issues related to safety, pharmacokinetics, and delivery. 

## Figures and Tables

**Figure 1 pharmaceutics-15-02766-f001:**
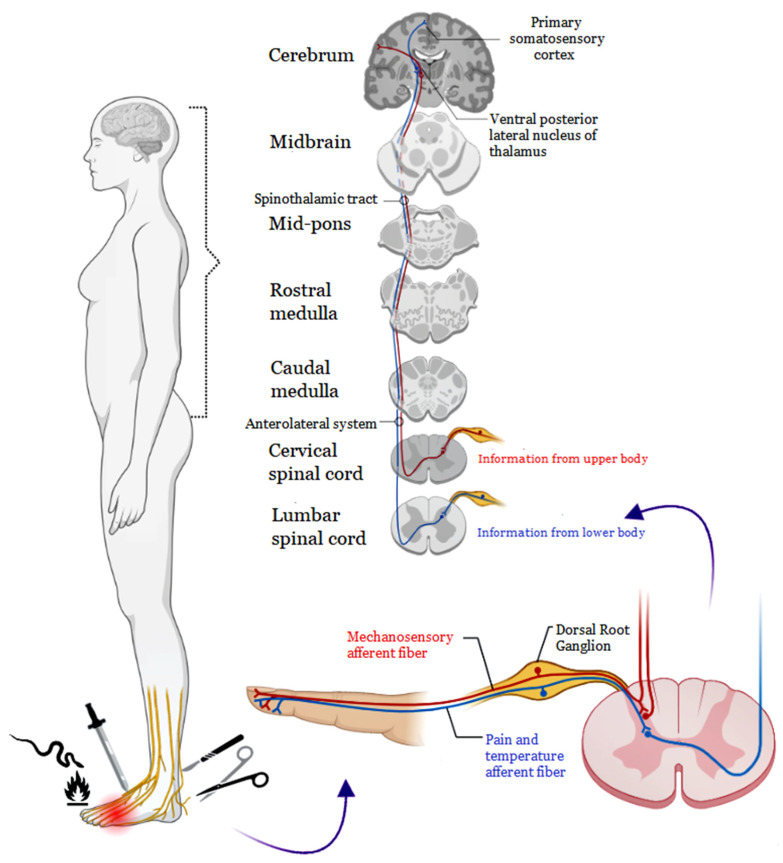
The process of detecting a noxious stimulus through nociceptors. Nociceptors feature a bifurcated projection: One branch extends as a peripheral axon, establishing connections with the skin and various organs, while the other branch projects toward the central nervous system (CNS). This intricate projection mechanism enables the transmission of information regarding the presence and intensity of the stimulus. The CNS then processes this information, orchestrating appropriate responses to potentially harmful stimuli.

**Figure 2 pharmaceutics-15-02766-f002:**
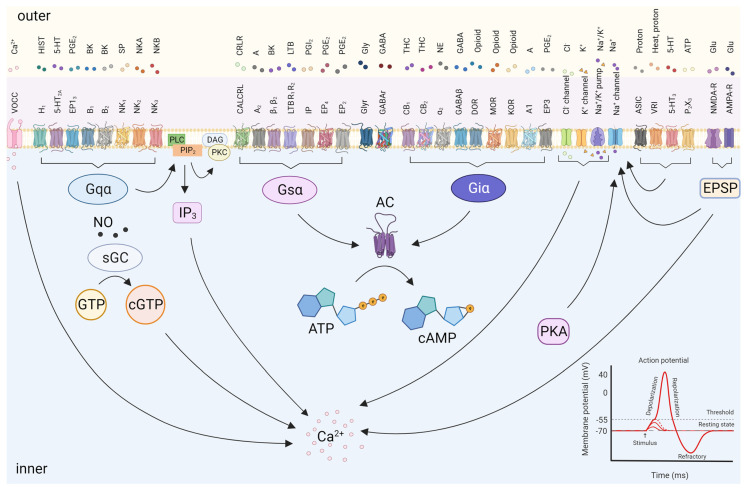
Neurotransmitters involved in inflammatory pain sensation as all the major types of neurotransmitters (inflammatory mediators and non-inflammatory mediators), second messenger production, and others. Abbreviations: 5-HT: 5-hydroxytryptamine; 5-HT2A: 5-hydroxytryptamine type 2A receptor; 5-HT3: 5-hydroxytryptamine type 3 receptor; A1: adenosine type 1 receptor; A2: adenosine type 2 receptor; AC: adenylyl cyclase; AMPA-R: amino-3-hydroxy-5-methyl-4-isoxazolepropionic acid receptors; ASIC: acid-sensing ion channels; ATP: adenosine triphosphate; B1: bradykinin receptor type B1; B2: bradykinin receptor type B2; BK: bradykinin; CALCRL: calcitonin receptor-like receptor; cAMP: cyclic adenosine monophosphate; CB1: cannabinoid type 1 receptors; CB2: cannabinoid type 2 receptors; cGMP: cyclic guanosine monophosphate; CGRP: calcitonin gene-related peptide; Cl^−^: chloride ion; DAG: diacylglycerol; DH: dorsal horn; DOR: δ-opioid receptors; EP: prostaglandin E2 receptor; EP1: prostaglandin E2 receptor type 1; EP2: prostaglandin E2 receptor type 2; EP3: prostaglandin E2 receptor type 3; EP4: prostaglandin E2 receptor type 4; EPSP: excitatory post-synaptic potentials; GABA: γ-aminobutyric acid; GABAA: γ-aminobutyric acid type A receptor; GABAB: γ-aminobutyric acid type B receptor; GlyR: glycine receptor; H1: histamine; IP: prostacyclin receptor; IP3: inositol triphosphate; K^+^: potassium ion; KOR: κ-opioid receptors; LTB4: leukotriene B4; LTB4-R1: leukotriene B4 type 1 receptor; LTB4-R2: leukotriene B4 type 2 receptor; Mg^2+^: magnesium ion; MOR: µ-opioid receptors; Na^+^: sodium ion; Nav: voltage-activated Na^+^ channels; NE: norepinephrine; NGF: nerve growth factor; NK1: neurokinin type 1 receptor; NK2: neurokinin type 2 receptor; NK3: neurokinin type 3 receptor; NKA: neurokinin A; NKB: neurokinin B; NMDA-R: N-methyl-D-aspartate receptors; NO: nitric oxide; P2X3: purino receptor; PAG: periaqueductal gray; PG: prostaglandins; PGE1: prostaglandin E1; PGE2: prostaglandin E2; PGI2: prostacyclin; PI3: phosphoinositide 3-kinase; PKA: protein kinase A; PKC: protein kinase C; PLC: phospholipase C; sGC: soluble guanylyl cyclase; SP: substance P; THC: tetrahydrocannabinol; VR1: vanilloid receptor for capsaicin; VSMC: vascular smooth muscle cell; α1: alpha 1-adrenoreceptor; α2: alpha 2-adrenoreceptor; β: beta-adrenoreceptor.

**Table 1 pharmaceutics-15-02766-t001:** Snake-derived toxins able to inhibit pain.

Toxin/ Molecule	Species	Production	Administration	Mechanism of Analgesia	Model	Ref.
µ-EPTX-Na1	*Naja atra*	Purified from venom	Intraplantar (i.pl.) injection	Nav1.8 channel inhibitor	Acute inflammation models induced by formalin and acetic acid, chronic inflammation pain model induced by complete Freund’s adjuvant and partial nerve ligation-induced allodynia.	[69]
Mambalgin-1	*Dendroaspis polylepis polylepis*	Purified from venom	Intravenous (i.v.) and intratechal (i.t.) injection	Acid-sensing ion channel (ASIC) inhibitors	Motor behavior tests such as accelerated rotarod test and grip strength test. Pain and inflammation models induced by carrageenan, thermal and mechanical pain test by von Frey	[68]
Mambalgin-3	*Dendroaspis polylepis polylepis*	Purified from venom	Intravenous (i.v.), intrathecal (i.t.), and intraplantar (i.pl.) injection	Acid-sensing ion channel (ASIC) inhibitors	Motor behavior tests such as accelerated rotarod test and grip strength test. Pain and inflammation models induced by carrageenan, thermal and mechanical pain test by von Frey	[68]
Cobra neurotoxin	*Naja naja atra*	Purified from venom	Intraperitoneal (i.p.) injection	Adenosine receptor (A1 and A2A) pathway activation	Acute pain model induced by hot plate and spinal cord injury	[74]
Najanalgesin	*Naja naja atra*	Purified from venom	Intrathecal (i.t.) injection	c-Jun N-terminal kinase (JNK) inhibitor	Neuropathic pain induced by spinal nerve ligation	[75]
Crotoxin (CTX)	*Crotalus durissus terrificus*	Purified from venom	Subcutaneous injection (s.c.)	Formyl Peptide, α2-Adrenergic and Muscarinic Receptors	Acute and chronic phases of hypernociception induced by partial sciatic nerve ligation	[70]
Crotoxin (CTX)	*Crotalus durissus terrificus*	Purified from venom	Subcutaneous injection (s.c.)	Analgesia dependent on formyl peptide, lipoxygenase and muscarinic receptors	Pain on the MOG 35-55-induced experimental autoimmune encephalomyelitis, an animal model of multiple sclerosis	[71]
Crotalphine	*Crotalus durissus terrificus*	Chemical synthesis	Oral administration (p.o.)	TRPA1	The ciguatoxin-induced cold hypersensitivity, and the bradykinin-induced and zymosan-induced mechanical hypersensitivity	[73]
Cobratoxin (CbTX)	*Naja naja kaouthia*	Purified from venom	Intrathecal (i.t.) injection	α7 nicotinic acetylcholine receptor (nAChRs)	Acute pain model induced by hot plate and tail-flick	[76]

**Table 2 pharmaceutics-15-02766-t002:** Scorpion-derived toxins able to inhibit pain.

Toxin/ Molecule	Species	Production	Administration	Mechanism of Analgesia	Model	Ref.
rAGAP	*Buthus martensii*	Recombinant	Injection application location not specified	CHis6-rAGAP and NHis6-rAGAP	Xenograft tumor mouse model	[78]
AGAP W38G	*Buthus martensii*	Purified from venom	Intraplantar (i.pl.) injection	Nav1.7 and Nav1.8 channel inhibitor	An acute thermal pain model induced by a hot plate and an acute inflammation model induced by formalin	[79]
BmK AGP-SYPU1	*Buthus martensii*	Recombinant	Intraperitoneal (i.p.) injection	Arginine residue at position 58	Mouse-twisting pain model	[85]
BmK AGAP	*Buthus martensii*	Recombinant	Intrathecal (i.t.) injection	MAPK inhibitor	An acute inflammation model induced by formalin, a thermal pain model induced by hot plate, and mechanical allodynia	[80]
BotAF	*Buthus occitanus tunetanus*	Purified from venom	Intraperitoneal (i.p.), intrathecal (i.t.) and intraplantar (i.pl.) injections	Ion channel inhibitor	An acute thermal pain model induced by a hot plate, a nociception model induced by a shutter-controlled lamp, and an inflammation model induced by formalin and acetic acid writhing	[81]
BmK AGAP	*Buthus martensii*	Purified from venom	Intraplantar (i.pl.) injection	Kv1.3 channel and MAPK inhibitor	An acute inflammation model induced by formalin	[82]
BmK AGAP	*Buthus martensii*	Recombinant	Intraperitoneal (i.p.) injection	MCF-7 and MDA-MB-231 migration inhibitor	Xenograft tumor mouse model	[77]
Makatoxin-3 (MkTxs)	*Buthus martensii*	Purified from venom	Intraperitoneal (i.p.) injection	Nav1.7 inhibitor	Acute nociception induced by formalin test and Freund’s adjuvant (CFA) induced mechanical pain model	[86]
TsNTxP	*Tityus serrulatus*	-	Intraperitoneal (i.p.) injection	Nav channel inhibitor	Acute nociception induced by a water bath and neuropathic pain model induced by CCI model	[83]
Leptucin	*Hemiscorpius lepturus*	Chemical synthesis	Intraperitoneal (i.p.) injection	Ion channel inhibitor	Acute thermal pain model induced by hot plate and nociception model using tail flick test	[84]

**Table 6 pharmaceutics-15-02766-t006:** Mollusk-derived toxins able to inhibit pain.

Toxin/ Molecule	Species	Production	Administration	Mechanism of Analgesia	Model	Ref.
Im10A (conotoxin)	*Conus imperialis*	Synthesized	Intramuscular injection (i.m.)	-	Rat PNL model	[131]
μ-Conotoxin TsIIIA	*Conus tessulatus*	Synthesized	Intrathecal (i.t.) injection	Inhibition of TTX (tetrodotoxin)-resistant sodium currents in DRG neurons	Hot-plate model	[132]
α-Conopeptide Eu1.6	*Conus eburneus*	Synthesized	Intramuscular injection (i.m.) and intravenous (i.v.) injection	Inhibition of high voltage-activated N-type calcium channel currents (Cav2.2) in isolated mouse dorsal root ganglia (DRG) neurons	Rat partial sciatic nerve injury (PNL model) and chronic constriction injury pain (CCI model) models	[102]
ω-Conotoxins: MoVIA and MoVIB	*Conus* *Moncuri*	Purified from venom and synthesized	Intrathecal (i.t.) injection	Inhibition of rat Cav2.2 channels	PNL-induced neuropathic pain	[134]
α-Conotoxin Vc1.1 Variants	-	Synthesized	Intramuscular (im.i), injection	Inhibition of G protein-coupled γ-aminobutyric acid type B receptors (GABABR) coupled Cav2.2 channels in rat DRG neurons	PNL and CCI models	[135]
cVc1.1 and cVc1.1 analogues: [C2H, C8F]cVc1.1 and [N9W]cVc1.1	-	Synthesized	Intra-colonic administration	Reduction in the excitability of DRG neurons	Chronic visceral hypersensitivity (CVH model)	[142]
ω -Conotoxin MVIIA modified	-	Synthesized	Intravenously (i.v.) injection and intranasally injection	-	Hot-Plate model	[144]
αO-Conotoxin GeXIVA	*Conus generalis*	Synthesized	Intramuscular (im.i), injection	-	Oxaliplatin-induced neuropathic pain, cold and mechanical allodynia	[137]
Partially purified conotoxins (C1-C7) of C. coronatus and C. frigidus (F1-F6).	*Conus coronatus* and *Conus frigidus*	Purified from Venom	Intraperitoneal (i.p.) injection	Only C2 had analgesic effects in both tested models; mechanism of analgesia: not studied	Hot-plate model and formalin-induced pain	[145]
BuIA conotoxin analogues	*Conus bullatus*	Synthesized	Intraperitoneal (i.p.) injection	All the analogs showed the same analgesic activity of BuIA Mechanism of analgesia: not studied	Hot-plate model and paclitaxel-induced neuropathic pain	[138]
ω-conotoxin Bu8	*Conus bullatus*	Synthesized	Intraperitoneal (i.p.) injection	Inhibition of Cav2.2	Hot-plate model and analgesic activity to acute pain and inflammatory pain	[146]
Conotoxin-Ac1// Conotoxin-Ac1-O6P (variant)	*Conus achatinus*	Synthesized	Intrathecal (i.t.) injection	Inhibition of NR2B ion channels	Hot-plate and tail-flick models	[141]
α-Conotoxin Vc1.1 modified	*Conus victoriae*	Purified from venom and modified	Intramuscular (im.i), injection	Activation of GABABRs expressed in DRG	Spinal nerve ligation-induced neuropathic pain (SNL model), mechanical allodynia	[136]
α4/7-Conotoxin (Lv1d)	*Conus lividus*	Synthesized	Intrathecal (i.t.) injection (mouse hot plate model) Intramedullary injection (formalin-induced pain model)	-	Mouse hot-plate and formalin-induced pain models	[147]
µ-Conotoxin S24a	*Conus striatus*	Synthesized	Intrathecal (i.t.) injection	-	Mouse hot-plate and formalin-induced pain models	[148]
ω-Conotoxin	*Conus virgo*	Purified from venom and modified	Intraperitoneal (i.p.) injection	-	Central analgesic assay (tail immersion test) and Peripheral analgesic assay (acetic acid-induced writhing test)	[149]
α-CTx RgIA analogues (RgIA-5524)	-	Synthesized	Subcutaneous (i.s.) injection	Inhibition of α9-containing nicotinic acetylcholine receptors (nAChRs)	Oxaliplatin-induced neuropathic pain	[143]
α-Conotoxin RgIA analogues (RgIA-5474)	-	Synthesized	Subcutaneous (i.s.) injection	-	Oxaliplatin-induced neuropathic pain, cold allodynia	[139]
ω–Conotoxins: MVIIA, GVIA and CVIF	-	Obtained from Alomone Labs (Jerusalem, Israel).	Intraplantar (i.pl.). injection	-	Post-surgical pain (PSP model) and cisplatin-induced pain neuropathy and oxaliplatin-induced neuropathic pain	[140]

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
