# Peer review of "Unveiling the Pain Relief Potential: Harnessing Analgesic Peptides from Animal Venoms"

_pharmaceutics, 2023, doi:10.3390/pharmaceutics15122766_

Round 1

Reviewer 1 Report

Comments and Suggestions for Authors

The authors present a review aims to highlight the most promising venom-derived toxins that have been investigates as potential prototypes for analgesic drugs. The manuscript is well-written and interesting. I have only minor comments.

Minor comments

#1. Figure 2 is missing.

#2. I do not understand following sentence “Snake venoms are true mutable weapons”. (Page 4, Line151)

#3. Please include generic name of Prialt (Page 18, Line 440).

Author Response

Dear Guest Editors and Reviewer,

Thank you very much for reviewing our manuscript. On behalf of all the authors, we also greatly appreciate the reviewers for their complimentary comments and suggestions. The following are our point-by-point responses.

# Reviewer 1

The authors present a review aims to highlight the most promising venom-derived toxins that have been investigates as potential prototypes for analgesic drugs. The manuscript is well-written and interesting. I have only minor comments.

Minor comments

#1. Figure 2 is missing.

Response: Thanks for the information. In fact, Figure 3 has been corrected to Figure 2.

#2. I do not understand following sentence “Snake venoms are true mutable weapons”. (Page 4, Line151)

Response: The sentence was rewritten: “Snake venoms are true mutable substances”.

#3. Please include generic name of Prialt (Page 18, Line 440). 

Response: Thanks for the suggestion. The generic name for Prialt has been included.

Reviewer 2 Report

Comments and Suggestions for Authors

The manuscript submitted for review is a thorough analysis of the cited works. Particularly noteworthy is the tabular presentation of the directions of action and absorption points of the discussed substances of animal origin. I agree with the authors on the advisability of searching for physiological and pathological phenomena in nature, such as pain and inflammation. The chemical diversity of the composition of animal venoms is noteworthy - the irritating substance is accompanied by compounds that also alleviate the effects of bites in many ways. As someone searching for molecules with potential pain-relieving properties, I expected to find inspiration for my work. Unfortunately, the mechanisms of understanding the phenomenon of pain were discussed without delving into the chemical characteristics of the discussed natural compounds isolated from venoms, which does not allow finding a relationship between their chemical structure and biological action. Of course, it is not easy to talk about large peptide molecules, but in the case of such a huge structure as vancomycin, it was possible to find fragments critical for action against resistant strains, which allows the structure of this antibiotic to be modified and its action to be extended to additional points of attachment and, consequently, to increase the scope of biological action. This is just an example, but the question arises whether the characteristic elements of the structure of active compounds are discussed in the literature cited in this manuscript.

A conclusion arises as to how many ways (mechanisms and points of action) natural substances influence the perception of pain. Perhaps this is one of the directions in which therapy should go, and it seems to be heading by recommending the composition of multimodal therapy in the fight against neuropathic pain. I did not find this conclusion in the summary of the manuscript, but it emerged after analyzing its content.

Author Response

Dear Guest Editors and Reviewer,

Thank you very much for reviewing our manuscript. On behalf of all the authors, we also greatly appreciate the reviewers for their complimentary comments and suggestions. The following are our point-by-point responses.

# Reviewer 2

The manuscript submitted for review is a thorough analysis of the cited works. Particularly noteworthy is the tabular presentation of the directions of action and absorption points of the discussed substances of animal origin. I agree with the authors on the advisability of searching for physiological and pathological phenomena in nature, such as pain and inflammation. The chemical diversity of the composition of animal venoms is noteworthy - the irritating substance is accompanied by compounds that also alleviate the effects of bites in many ways. As someone searching for molecules with potential pain-relieving properties, I expected to find inspiration for my work. Unfortunately, the mechanisms of understanding the phenomenon of pain were discussed without delving into the chemical characteristics of the discussed natural compounds isolated from venoms, which does not allow finding a relationship between their chemical structure and biological action. Of course, it is not easy to talk about large peptide molecules, but in the case of such a huge structure as vancomycin, it was possible to find fragments critical for action against resistant strains, which allows the structure of this antibiotic to be modified and its action to be extended to additional points of attachment and, consequently, to increase the scope of biological action. This is just an example, but the question arises whether the characteristic elements of the structure of active compounds are discussed in the literature cited in this manuscript.

Response: The study's primary focus was never to provide an in-depth exploration of pain mechanisms, as these mechanisms are complex, involving multiple pathways and modes of action. We appreciate your suggestion and will consider it for inclusion in our group's future projects.

A conclusion arises as to how many ways (mechanisms and points of action) natural substances influence the perception of pain. Perhaps this is one of the directions in which therapy should go, and it seems to be heading by recommending the composition of multimodal therapy in the fight against neuropathic pain. I did not find this conclusion in the summary of the manuscript, but it emerged after analyzing its content.

Response: Thanks for the suggestion. The conclusion was rewritten considering the potential for multimodal therapy, and the conclusion has been included in the summary.

Reviewer 3 Report

Comments and Suggestions for Authors

In this review, the authors focus on various venom-derived toxins that could be used as potential prototypes for analgesic drugs. It is interesting and I think that this study is meaningful as researchers interested in the field of pain could have a glimpse on various potential analgesic drug. I have only few comments:

Although in the beginning of the manuscript, authors have assessed the difference mechanisms of pain, it appears that the different mechanisms are not connected with the different analgesic mechanism of toxin-derived agents. They seem to be presented as two things apart. It will be more meaningful if authors could connect these two in the paper. 

Also there should be a section where the toxic or side effects of agents are presented. This could be one of the most important point as all components are toxin-derived.  

The quality of the figures are not good, especially figure 1. I suggest that authors replace it to a higher resolution one. 

Author Response

Dear Guest Editors and Reviewer,

Thank you very much for reviewing our manuscript. On behalf of all the authors, we also greatly appreciate the reviewers for their complimentary comments and suggestions. The following are our point-by-point responses.

# Reviewer 3

In this review, the authors focus on various venom-derived toxins that could be used as potential prototypes for analgesic drugs. It is interesting and I think that this study is meaningful as researchers interested in the field of pain could have a glimpse on various potential analgesic drug. I have only few comments:

Although in the beginning of the manuscript, authors have assessed the difference mechanisms of pain, it appears that the different mechanisms are not connected with the different analgesic mechanism of toxin-derived agents. They seem to be presented as two things apart. It will be more meaningful if authors could connect these two in the paper.

Response: The objectives of this work did not include an in-depth discussion of pain mechanisms; it is a complex and multifactorial mechanism. We appreciate your suggestion; however, though, and plan to incorporate this subject in our forthcoming research.

Also there should be a section where the toxic or side effects of agents are presented. This could be one of the most important points as all components are toxin-derived. 

Response: Thanks for your contribution. In fact, this topic holds significant relevance. However, our current screening process restricts us to elucidating the analgesic effects of the peptides, since apparently, as far as we know, work has not yet progressed towards the toxicological characterization of these molecules. We still do not have or have few peptide drugs based on analgesia and the effects of the toxins themselves are a great challenge for the construction of these drugs, such as melittin, which has action on several receptors and has toxic and harmful actions.

The quality of the figures are not good, especially figure 1. I suggest that authors replace it to a higher resolution one.

Response: The quality of the figures has been enhanced with higher resolution.

Reviewer 4 Report

Comments and Suggestions for Authors

The revision article by Marques Pereira et al. go over peptides extracted from animal venoms and showing beneficial activity in different preclinical model of pain. Although having all these peptide toxins, they mechanism of action and the different model of pain linked in a unique compendium is very attractive, the manuscript needs deep amendment before publication.
Sentences are often very long and convoluted, which complicated the comprehension, i.e L223-226, 291-293, 295-297, etc.
Regarding the content, the manuscript could benefit by including a few clinical trials (NCT01491321, NCT00949754 and NCT01112722) on bee venom and Apitox for low back pain and osteoarthritis. These could serve to enhance the interest in these toxins.
Similarly, in the case of frog-derived toxins, the manuscript lacks for some comment on the potent analgesic peptides extracted from frog skin, like Dermorfphin, Deltorphin and Analgesin.
Sometimes there is more information in the tables that within the text related to the tables, which turns out to be a simple enumeration of the table's contents. Some more details/information from the original papers would be expected within the text, especially on aspects that are not in the tables. In addition, tables should be lightened by using abbreviations for targets and pain models, avoiding long sentences explaining mechanisms of animal models, and should be ordered alphabetically by the toxin name.
In future perspectives, the current improvements in peptide delivery systems should be included.
Table 1 headings (also apply to the following ones): Use “Way of Administration or Administration” instead “Formulation”, the meaning is totally different in this context.
Table 1: mambalgin-1 and 3, last column, I suppose that thermal means “hot-plate test”, the Von Frey test is only for mechanical allodynia. “zymosaninduced” should be “zymosan-induced” (last column, right)
All tables: Abbreviation for intraperitoneal is “i.p.”; subcutaneous “s.c.”; and oral “p.o.”
The neuropathic pain model induced by constriction of a sciatic nerve, is commonly abbreviated as CCI model
Epibatidine is not a peptide, so please remove L373-374, and the corresponding line in the table, and reference.
Other minor errors:
L201: “for spiders for” it is supposed to be “for spiders” (remove second “for”)
L312: “all belonging”
L316:”these” should be “These” (capital letter)
L323:  “water bath”, means tail-flick test?
L329 (and table 2): “arginine residue at position 58” is not a target, please revise the reference to identify the target/mechanism by which Bmk AGP-SYPU1 induced the antinociceptibe effect.
Page 10, Makatoxin-3: “reund's adjuvant” should be “Freund's adjuvant”. Remove  (Sigma-Aldrich, USA)
L253: remove “.” Before “new”
L266: consider to change “in” with “of”
L294: consider to change “by” by “through”
L350:: “with” should be “of”
L369: the sentence seems incomplete
L449: consider to change “mass” with “molecular weight”

Reference 41: All Capital letters??
References: For journal names, a mix of abbreviated and complete names has been used.

Comments on the Quality of English Language

Deep English editing is needed and grammar punctuation should be revised all along the text.

Author Response

# Reviewer 4

The revision article by Marques Pereira et al. go over peptides extracted from animal venoms and showing beneficial activity in different preclinical model of pain. Although having all these peptide toxins, they mechanism of action and the different model of pain linked in a unique compendium is very attractive, the manuscript needs deep amendment before publication.

Sentences are often very long and convoluted, which complicated the comprehension, i.e L223-226, 291-293, 295-297, etc.

Regarding the content, the manuscript could benefit by including a few clinical trials (NCT01491321, NCT00949754 and NCT01112722) on bee venom and Apitox for low back pain and osteoarthritis. These could serve to enhance the interest in these toxins.

Response: We appreciate your suggestion. However, as this work presents a scope of peptides that can be translated into the clinic, including information on toxins currently undergoing clinical trials falls outside the objectives of the present study.

Similarly, in the case of frog-derived toxins, the manuscript lacks for some comment on the potent analgesic peptides extracted from frog skin, like Dermorfphin, Deltorphin and Analgesin.

Response: Our search methodology aimed to identify current and original articles discussing peptides with analgesic potential. Consequently, peptides derived from frog skin, which were documented in publications dating back to the late 1990s and early 2000s, were excluded from the screening process.

A paragraph about peptide delivery system has been added to the “Future perspectives” section.

Table 1 headings (also apply to the following ones): Use “Way of Administration or Administration” instead “Formulation”, the meaning is totally different in this context.

Response: The term "Formulation" has been replaced with "Administration" in all tables.

Table 1: mambalgin-1 and 3, last column, I suppose that thermal means “hot-plate test”, the Von Frey test is only for mechanical allodynia. “zymosaninduced” should be “zymosan-induced” (last column, right).

Response: The term "zymosaninduce" was corrected to "zymosan-induced."
